# Temporal Resistome and Microbial Community Dynamics in an Intensive Aquaculture Facility with Prophylactic Antimicrobial Treatment

**DOI:** 10.3390/microorganisms8121984

**Published:** 2020-12-13

**Authors:** Hemant J. Patil, Joao Gatica, Avihai Zolti, Ayana Benet-Perelberg, Alon Naor, Barak Dror, Ashraf Al Ashhab, Sophi Marman, Nur A. Hasan, Rita R. Colwell, Daniel Sher, Dror Minz, Eddie Cytryn

**Affiliations:** 1Institute of Soil, Water and Environmental Sciences, Volcani Center, Agricultural Research Organization, P.O. Box 15159, Rishon LeZion 7528809, Israel; hemantpatil8311@gmail.com (H.J.P.); joaogatica@hotmail.com (J.G.); avihai.zolti@mail.huji.ac.il (A.Z.); barakd@volcani.agri.gov.il (B.D.); minz@volcani.agri.gov.il (D.M.); 2Department of Plant Pathology and Microbiology, The Robert H. Smith Faculty of Agriculture, Food and Environment, The Hebrew University of Jerusalem, Rehovot 91905, Israel; 3Dor Aquaculture Research Station, Fisheries Department, Israel Ministry of Agriculture and Rural Development, Dor 3082000, Israel; ayanab@moag.gov.il (A.B.-P.); Alonn@moag.gov.il (A.N.); 4The Dead Sea and Arava Science Center, Masada 86900, Israel; ashraf.ashhab@gmail.com; 5Department of Marine Biology, Leon H. Charney School of Marine Sciences, University of Haifa, Haifa 3498838, Israel; sofimarman@gmail.com (S.M.); dsher@univ.haifa.ac.il (D.S.); 6CosmosID Inc., Rockville, MD 20742, USA; nur.hasan@cosmosid.com (N.A.H.); rcolwell@umiacs.umd.edu (R.R.C.); 7Center for Bioinformatics and Computational Biology, University of Maryland, College Park, MD 20742, USA; 8Maryland Pathogen Research Institute, University of Maryland, College Park, MD 20742, USA

**Keywords:** antimicrobials, antimicrobial resistance, aquaculture, metagenome, microbiome, resistome

## Abstract

Excessive use of antimicrobials in aquaculture is concerning, given possible environmental ramifications and the potential contribution to the spread of antimicrobial resistance (AR). In this study, we explored seasonal abundance of antimicrobial resistance genes and bacterial community composition in the water column of an intensive aquaculture pond stocked with Silver Carp (*Hypophthalmichthys molitrix*) prophylactically treated with sulfamethoprim (25% sulfadiazine; 5% trimethoprim), relative to an adjacent unstocked reservoir. Bacterial community composition was monitored using high-throughput sequencing of 16S *rRNA* gene amplicons in eight sampling profiles to determine seasonal dynamics, representing principal stages in the fish fattening cycle. In tandem, qPCR was applied to assess relative abundance of selected antimicrobial resistance genes (*sul1, sul2, dfrA1, tetA* and *blaTEM*) and class-1 integrons (*int1*). Concomitantly, resistomes were extrapolated from shotgun metagenomes in representative profiles. Analyses revealed increased relative abundance of sulfonamide and tetracycline resistance genes in fishpond-03, relative to pre-stocking and reservoir levels, whereas no significant differences were observed for genes encoding resistance to antimicrobials that were not used in the fishpond-03. Seasons strongly dictated bacterial community composition, with high abundance of cyanobacteria in summer and increased relative abundance of *Flavobacterium* in the winter. Our results indicate that prophylactic use of sulfonamides in intensive aquaculture ponds facilitates resistance suggesting that prophylactic use of these antimicrobials in aquaculture should be restricted.

## 1. Introduction

The global spread of antimicrobial resistance (AMR) has severe implications for human health [1]. For example, approximately 70% of the agents causing hospital acquired infections in the United States are resistant to at least one antimicrobial family [2], demonstrating the severe public health risk associated with AMR [3]. The dynamics of environmental AMR are currently not well understood, but there is evidence that anthropogenic activities such as animal husbandry, aquafarming and wastewater treatment may contribute to this phenomenon [4,5,6]. AMR can propagate across microbial communities via horizontal gene transfer (HGT) through mobile genetic elements such as plasmids, transposons and integrons [7,8]. As a result, anthropogenic- and environmentally-derived bacteria can exchange antimicrobial resistance genes (ARGs), and these HGT events can be maintained under selective pressure.

The use of antimicrobial compounds in aquaculture has become a common practice throughout the world [9,10,11,12,13]. Although clinical and prophylactic application of antimicrobials have proven valuable in preventing and treating fish disease, excessive application of antimicrobials in this rapidly growing industry may facilitate increased abundance of antimicrobial resistant bacteria (ARB) and antimicrobial resistance genes (ARGs) [11]. ARB and ARGs from aquaculture can disseminate to food webs and enter the water cycle, and alter microbial community structure. Furthermore, ARGs harboring mobile genetic elements can mobilize to clinically relevant bacteria resulting into global propagation of AMR (FAO/OIE/WHO, 2006) [14]. Nakayama et al. [13] reported the ubiquitous presence of sulfonamide resistance genes coupled with remarkably low bacterial diversity at interfaces of wastewater streams from backyard-based aquaculture units in Vietnam, associated with high load of sulfonamide antimicrobials used in those systems. Tang et al. [15] suggested temperature as a key driver of the bacterial community structure in freshwater aquaculture systems. Moreover, other studies have provided valuable information about AMR in relation to microbial community composition and temporal changes. Despite several past studies, comprehensive assessment of AMR dynamics in conventional intensive aquaculture that apply antimicrobials has not been investigated.

Recently, shotgun metagenomics has contributed significantly to microbial ecology by describing structure (who is there?) and function (what are they doing?) of complex environmental microbial communities [16]. In contrast to conventional 16S *rRNA* gene-amplicon sequencing, shotgun genomics adds the capacity to classify and quantify specific microbial genes in real time as a function of environmental parameters, thereby elucidating associations among specific microbial communities and the host genotype/phenotypes comprising the community [17,18,19]. For example, metagenomics was applied to detect co-occurrence of specific ARGs on both viromes and bacterial genomes in an experimental aquaculture facility, suggesting that transduction of ARGs takes place in these systems [20].

In a previous culture-based study targeting *Aeromonas* in the water columns of two commercial-scale aquaculture facilities, we demonstrated that sulfonamide and tetracycline gene abundance were positively correlated to sulfadiazine/trimethoprim and oxytetracycline that were applied prophylactically and clinically, respectively. In contrast, these isolates displayed lower resistance to antimicrobials (chloramphenicol, ceftriaxone and gentamicin) that were not used [12]. The aim of this study was to understand the structural and functional dynamics of the resistome and microbiome in the water column of an intensive aquaculture system employing antimicrobials prophylactically. Standard water monitoring methods were integrated with advanced culture-independent molecular and metagenomic platforms.

## 2. Materials and Methods

### 2.1. Sampling and DNA Extraction

Water samples for 16S*rRNA* gene amplicon and qPCR analysis were collected from fishpond-03 (FP-03) and reservoir (Res) of the Dor aquaculture research station, Israel, details of which have previously been described [12] and sample wise analysis details were tabulated below (Appendix A). Altogether, eight profiles were sampled for the duration of a Silver Carp (*Hypophthalmichthys molitrix*) fattening cycle: 6 July 2015; 4 August 2015; 18 August 2015; 31 August 2015; 14 October 2015; 22 December 2015, 10 February and 9 March 2016. All samples were collected in triplicate (except 6 July, 4 August, and 18 August, when for logistical reasons samples were collected in duplicate and 10 February where only a single replicate was sampled). Samples were immediately filtered (25–30 mL depending on turbidity) onto glass microfiber filters (Whatman GF/F; 0.7µ; dia 25 mm) for nutrient analyses, and subsequently onto 0.22µ polycarbonate filters for DNA extraction and subsequent molecular analyses. Filters were preserved in conservation buffer (50 mM Tris, pH = 8.3, 0.75 M Sucrose, 40 mM EDTA), transferred on ice to the laboratory within 2 h and stored at −80 °C until DNA extraction. Subsequently, frozen filters with conservation buffer added to the tubes were transferred on ice to the Biomedical Core Facility of the Technion for robotic DNA extraction. The first step of extraction included chemical and mechanical extraction as follows: filters were thawed, centrifuged for 10 min at 15,000× *g* and the conservation buffer removed. Next, Blood and Tissue Kit lysis buffer (Qiagen) was added, and the samples were mechanically extracted using two 3 mm stainless steel beads and a speed of 30/s for 1.5 min on the TissueLyser LT (Qiagen). An amount of 30 µL of lysozyme was added and the tubes incubated at 37 °C for 30 m, after which 25 µL and 200 µL proteinase K and buffer AL (Qiagen) were added, respectively, and the tubes incubated at 56 °C for 1 h on a shaker. Finally, the tubes were centrifuged for 10 min at 5000× *g* and the upper liquid was transferred to a new 2 mL Eppendorf tube for further extraction using the Qiacube robot (Qiagen) and the DNeasy Blood and Tissue Kit. Extracted DNA was maintained at −20 °C serving as working stock, and at −80 °C as preserved stock for 16S rRNA gene amplicon sequencing and shotgun metagenomic analyses described below.

### 2.2. Physiochemical Analyses

Analysis of ammonia, nitrite and nitrate was performed using an autoanalyzer continuous flow system (Lachat QuikChem^®^ 8500 Series 2 Flow Injection Analyzer, Milwaukee, WI, USA). The pH was measured with a pH meter (EUTECH Instruments pH 700), while average temperature and precipitation at the target site during the study period was obtained from the Israel Meterological Service database (https://ims.data.gov.il/).

### 2.3. Sequencing of 16S rRNA Gene Amplicons

All 38 (19 FP-03 and19 Res) environmental DNA samples were amplified separately targeting the V3-V4 *16S rRNA* gene region using primer set CS1_341F and CS2_816R [21] with a two-stage “targeted amplicon sequencing (TAS)” protocol [22,23]. The primers contained 5′ common sequence tags (known as common sequence 1 and 2 or CS1 and CS2) as described previously by Moonsamy et al. [24]. The PCR reaction was conducted in a final volume of 25 µL and PCR conditions were as follows: 1 cycle of 95 °C for 5 min, followed by 28 cycles of 95 °C for 30 s, 50 °C for 30 s and 72 °C for 60 s with final elongation of 72 °C for 5 min. Subsequently, sond PCR amplification was performed in 10µl reactions in 96-well plates. A mastermix for the entire plate was prepared using MyTaq HS 2X mastermix. To each well, a separate primer pair with a unique 10-base barcode was added, obtained from the Access Array Barcode Library for Illumina (Fluidigm, South San Francisco, CA; Item# 100-4876). These Access Array primers contained CS1 and CS2 linkers at the 3′ ends of the oligonucleotides. Cycling conditions were as follows: 95 °C for 5 min, followed by 8 cycles of 95 °C for 30 s, 60 °C for 30 s and 72 °C for 30 s. A final, 7-min elongation step was performed at 72 °C. PCR products were purified using SequalPrep plates (Life Technologies) according to manufacturer’s instructions. Subsequently, the PCR products were quantified using a Quant-iT PicoGreen dsDNA Assay Kit (Thermo Fisher) implemented on a Genios Pro Fluorescence microplate reader (Tecan). PCR products were pooled using PicoGreen quantification on an epMotion5075 liquid handling workstation (Eppendorf).

The pooled libraries, along with a 15% phiX spike-in, were loaded to a MiSeq v3 flow cell, and sequenced using an Illumina MiSeq sequencer. Fluidigm sequencing primers, targeting CS1 and CS2 linker regions, were used to initiate sequencing. De-multiplexing of reads was performed on the instrument. Library preparation and pooling was performed at the DNA Services (DNAS) facility, Research Resources Center (RRC), University of Illinois at Chicago (UIC). Sequencing was performed at the W.M. Keck Center for Comparative and Functional Genomics at the University of Illinois at Urbana-Champaign (UIUC). Raw sequences were processed through downstream analysis, using the below mentioned bioinformatic tools.

### 2.4. Metagenome Analysis Employing Shotgun Sequencing

Extracted DNA was prepared and processed for sequencing using Nextera XT kit (Illumina, San Diego, CA, USA) according to the manufacturer’s instructions, and sequenced at the Genome Research Division Sequencing Core, University of Illinois, Chicago, USA. After processing, libraries were assessed for size using an Agilent TapeStation 2000 automated electrophoresis device (Agilent Technologies, Santa Clara, CA, USA) and for concentration by a Qubit flurometer (Thermo Fisher Scientific Inc., Waltham, MA, USA). Libraries were pooled in equimolar ratio and sequenced on a mid-output kit using an Illumina NextSeq500 sequencer, with paired-end 2 × 150 base reads.

### 2.5. Bioinformatic Analysis

#### 2.5.1. 16S *rRNA* Gene Amplicon Analysis

Amplicon sequences were subjected to an initial quality control step followed by bioinformatic analysis using a pipeline that integrated tools from QIIME v.1.91 [25] and MOTHUR v.1.33.3 [26]. Briefly, sequences containing more than one ambiguous base, those having a homopolymer length longer than 8bp, and sequences with an average quality score below 25 were removed from analysis. We included fragments ranging from 440 to 475 bp after adjacent PCR primer prioritizing quality of the sequences. Sequences were aligned using the SILVA reference database [27] and potential chimeric sequences were detected and removed using the chimera.uchime of MOTHUR. Sequences were assigned to operational taxonomic units (OTUs), based on ≥97% sequence similarity cutoff. Alpha diversity of the samples was calculated using the Simpson diversity index, while the Bray–Curtis dissimilarity matrix, which incorporates both membership and abundance, was used to interpret beta diversity among the samples and was calculated using the QIIME and MOTHUR combined pipeline.

#### 2.5.2. Shotgun Metagenome Analysis

Four selectedFP-03 (July 2015; August 2015; December 2015, and March 2016) and four previously sampled reservoir/Res (July 2013; August 2013; January 2014, and March 2014) profiles were selected for shotgun metagenome analysis. Two annotation pipelines were used to interpret the shotgun metagenomics data. The COSMOSID pipeline (app.cosmosid.com; [28,29,30]) was used to elucidate microbial (bacteria, virus, fungus and protist) community composition of the FP-03 water column samples, identify specific potential pathogens and pinpoint dominant ARGs in the water column samples. This pipeline facilitates rapid identification and quantifies microbial species, even from unassembled short NGS reads. A non-automated custom pipeline was also employed in the analysis for characterizing ARGs in the targeted samples. For the latter pipeline, raw reads were quality checked using FastQC, followed by trimmomatic [31]. BLAST analysis was performed using a downloaded version (1.1.3) of the Comprehensive Antibiotic Resistance Database (CARD). All reads were annotated and identified as ARG, according to its best BLAST hit [32], with minor modifications as follows: BLASTx with E-value cut-off set at ≤10^−6^, considering ≥80% similarity and ≥30 bit score to establish the ARG profile as well as abundance. The metagenomic data were deposited in the NCBI Sequence Read Archive under accession number SRP101485.

### 2.6. Quantification of ARGs Using qPCR

Amongst extracted DNA, eight samples (one each from all sampling profile as detailed in Appendix A) of FP-03 were prepared as described above, and subjected to quantitative PCR amplification. The analysis was targeting five ARGs: *sul1, sul2, dfrA1, tetA* and *blaTEM;* the class-1 integron integrase gene *intI1*; and the *16S rRNA* gene (which provides a rough estimate of total bacterial abundance in a given sample). Fast Real Time PCR was employed (Applied Biosystems). PCR was conducted using a final volume of 20 μL with master mix reagents as follows. Amplification programs were: 1 cycle of 95 °C for 10 min, 40 cycles of 95 °C for 15 s and 60 °C for 60 s with POWER and SELECT master mix reagents (Thermo Scientific), following 1 cycle of 95 °C for 5 min, 40 cycles of 95 °C for 5 s and 60 °C for 30 s with FAST SYBR green reaction master mix reagents (Thermo Scientific). Standards for respective genes were in a tenfold dilution series (gene copy numbers of 1 × 10^1^, 1 × 10^2^, 1 × 10^3^, 1 × 10^4^, 1 × 10^5^ and 1 × 10^6^/reaction) run with prepared Dor FP-03DNA samples. Each sample was analyzed using three technical replicates for each run. Primer details are provided in Table 1. Quantitative analysis of unknown samples was performed using standard curves generated from the amplification plot of known concentrations of the respective standard [33]. Efficiencies of all qPCR reactions ranged from 95% to 105%, except for *sul1*. Limit of detection for all of the screened ARGs was 10 copies.

### 2.7. Statistical Analyses

Statistical analyses were performed to determine correlations between abiotic factors, such as temperature, pH and average precipitation, with relative abundance of ARGs estimates made by qPCR and the bacterial communities delineated as OTUs at phylum as well as genus level. Pearson Correlation Coefficient was calculated to reflect variance among the datasets with respect to significance. Similarly, Canonical Correspondence Analysis (CCA) was performed to determine relative impact among the above-mentioned factors on each other. The relative abundance of ARGs was transformed to log (base 10) for ease of analysis. All statistical analyses were performed by PAleontologicalSTatistics software (PAST, version 3.08). Heatmaps were generated using the Python 3.1 Seaborn data visualization package. PERMANOVA test was conducted by “vegan” package in “R” [39].

The 16S rRNA gene amplicon sequencing data from nineteen FP-03 and Res samples and the shotgun metagenomic data from the eight FP-03 and Res samples were collectively deposited to the NCBI Bioproject repository as project # PRJNA375891entitled “Temporal analysis of intensive aquaculture water column”.

## 3. Results

### 3.1. Nutrient Analysis of Water Samples

Variable levels of ammonia (0.05–0.96 mg/L), nitrite (0.01–0.19 mg/L) and nitrate (0.04–5.73 mg/L) were observed in both FP-03 and Res water columns (Appendix A, Appendix A). Despite significantly higher fish densities, ammonia and nitrate levels measured in FP-03 were similar or only slightly higher than those measured in the Res.

Precipitation, temperature and pH profiles for samples collected in this study are shown in Appendix A. Rainfall was highest in February 2016 (63.7 mm), and there was no rainfall in July and August, 2015. Maximum and minimum temperatures were recorded in August 2015 (27.8 °C) and December 2015 (14.7 °C), respectively. The pH of the water samples ranged from 7.7 to 8.2.

### 3.2. Microbial Community Analysis

Evaluation of bacterial community structure in the FP-03 water column at eight selected time points during the fish fattening cycle was achieved by targeting 16S rRNA gene amplicon sequences. A total of 2,429,487 Illumina MiSeq-generated reads were obtained for 19 FP-03 and 19 Res samples in eight temporal profiles. Each sample contained between 40,000 and 160,000 reads with an average of 51,600 good quality reads per sample. Alpha diversity of the bacterial communities in the FP-03 and Res water columns at various sampling times was estimated using the Simpson diversity index and species richness (Figure 1). In general, we did not observe statistically significant differences in diversity for water column samples. However, the diversity of FP-03 was generally lower in the summer and relatively higher in the fall and winter. In the reservoir, lower diversity and species richness was observed in the summer and winter, respectively, relative to the same seasonal profiles in FP-03.

We applied non-metric multidimensional scaling (NMDS) to observe dissimilarities in microbial community composition among all the sampling profiles (Figure 2). Samples were grouped based on date of sampling (PERMANOVA of Bray–Curtis dissimilarity matrix, F = 8.59, *p* value = 0.001) and specifically segregated based on seasonality (summer and autumn—from July to October, versus winter and spring—December to March), at the first NMDS axis. This seasonal trend is supported by shifts in specific bacterial taxa described in the paragraph below. Although seasonality was the stronger driver, we also observed a statistically significant difference between the FP-03 vs. Res samples (FP vs. RES, F = 2.866, *p* value = 0.017).

At the phylum level, *Proteobacteria*, *Actinobacteria*, *Bacteriodetes* [40] and *Verrucomicrobia* were abundant in the water columns of both the FP-03 and Res, and *Plantomycetes* was also present at lower levels (Appendix A). *Cyanobacteria* were abundant in the summer profiles, but almost completely absent in the winter. Genus level taxonomical analysis revealed that these were primarily *Microcystis*, although *Synechococcus* were present in several Res profiles, inversely correlated to the relative abundance of *Microcystis* (Figure 3). The facultative phototrophic purple sulfur bacterium *Rhodobacter* was abundant in both water column samples throughout the year, as was the metabolically versatile genus *Sphingomonas*. *Pseudomonas* spp. was ubiquitous in both water column systems, but apparently inversely correlated with *Microcystis* blooms. *Novosphingobium*, *Bradyrhizobium* and *Fluviicola* spp. were present in all samples, but at lower relative abundances. *Flavobacterium* spp. were abundant in winter in both water column microbiomes, but almost undetectable in summer.

To identify specific clinically-associated bacteria, eukaryotes and viruses in the FP-03 and Res water columns, as mentioned above in the materials and methods section, four FP-03 (July 2015; August 2015; December 2015, and March 2016) and four previously sampled Res (July 2013; August 2013; January 2014, and March 2014) samples were analyzed by shotgun metagenome analysis using the COSMOSID pipeline. Abundance of human/animal-associated bacterial pathogens and commensals (Figure 4) included *Mycobacterium, Aeromonas, Elizabethkingia* and *Enterobacter* spp., as well as *Escherichia coli* and *Staphylococcus* spp. With the exception of *Mycobacterium* UM WGJ (isolated from a suspected tuberculosis patient in Malaysia) of these potential pathogens comprised less than 1% of the total bacterial community. Most of these were unique to a specific profile, suggesting these bacteria represent transient components of the water column microbiome.

COSMOSID bioinformatics were further applied to assess composition of viruses (Appendix A), protists (Appendix A) and virulence genes in the eight targeted FP-03 and Res water column samples. Almost all of the viruses that were detected were bacteriophages and the vast majority were *Microcystis* phages, corresponding to the *Microcystis* blooms observed in the summer in both the FP-03 and Res. Bacteriophages associated with *Pelagibacter* and *Enterobacteriacae* also were ubiquitous in most of the samples. Protist diversity was significantly higher in FP-03 water. While *Thalassiosira*, *Pseudoperonospora cubensis* and *Paramecium biaurelia* were ubiquitous in water at both sites, *Acanthamoeba palestinensis*, *Hammondia hammondi, Reticulomyxa filosa* and *Acanthamoeba mauritaniensis* were highly abundant and unique in the FP-03. A myriad of virulence genes associated with the *Klebsiella pneumonia*, *Salmonella typhimurium*, *Pseudomonas aeruginosa*, *Escherichia coli* and *Proteus mirabilis* were detected in all of the samples, but were for the most part substantially more abundant in the FP-03 than in the Res samples.

### 3.3. Assessment of ARGs Using Shotgun Metagenomics and qPCR

Relative abundance of ARGs in FP-03 and Res water was assessed using shotgun metagenomics and analyzed with both the COSMOSID and the CARD bioinformatic platforms, respectively. Subsequently, quantitative estimates of selected clinically relevant ARGs in the FP-03 water samples were conducted using qPCR.

A large fraction of the identified ARGs (Appendix A) encode for efflux pumps, which are often intrinsic and generally not associated with mobile genetic elements. We, therefore, focused on genes that can be potentially mobile. The most abundant of these ARGs identified using the COSMOSID and CARD bioinformatic pipelines are shown in the heatmaps in Figure 5A,B, respectively. Collectively, more ARGs were detected in the FP-03 compared to the Res water column samples. Sulfonamide resistance genes (*sul1* and *sul2*) were ubiquitous in the FP-03 samples, but were almost completely absent in the Res samples. While *sul2* was present in all of the profiles including July 2015, prior to antimicrobial administration, *sul1* was most abundant subsequent to the prophylactic treatment of fish with trimethoprim-sulfamethoxazole. In contrast, trimethoprim (administered prophylactically, along with sulfamethoxazole) resistance genes were not correlated with antimicrobial use and a substantial level of discrepancy in the detection of trimethoprim resistance genes occurred between the applied bioinformatic pipelines. COSMOSID analysis identified trimethoprim resistance genes *dfrA31* and *dfrD*, but only in the July 2013 Res samples, whereas CARD analysis showed high relative abundance of *dfrE* in all of the FP-03 and Res water samples. Based on COSMOSID analyses, tetracycline resistance genes were substantially more abundant in the FP-03 (with the exception of *tetG* in the March 2014 Res profile). This was, however, not the case in the CARD analysis, where a substantially higher number of tetracycline resistance genes was identified. The β-lactamase gene *bla*_OXA_ was detected in all samples by CARD analyses, but only in a few samples by COSMOSID analysis and no substantial differences were observed between the FP-03 and Res samples. Collectively, these analyses demonstrate that different bioinformatic platforms can produce different results on the same data. However, both platforms clearly indicated profuse and ubiquitous presence of sulphonamide resistance genes (*sul*1 and *sul*2) in the FP-03 and almost complete absence of these genes in the Res. This suggests a correlation between these genes and the prophylactic antimicrobial use in the FP-03.

We subsequently applied quantitative PCR (qPCR) to assess the relative abundance of selected ARGs in water samples from FP-03 in eight profiles spanning the duration of the fish fattening cycle from July 2015 to March 2016 (Figure 6). Efficiencies of all of the qPCR reactions ranged from 95% to 105%, except for *sul*1. Limit of detection for all ARGs was 10 copies. The abundance of *sul1* (ranging from 6.8 × 10^4^ to 3.5 × 10^5^ copies mL^−1^) substantially fluctuated during the fish fattening cycle, whereas *sul2* abundance was extremely low in the pre-stocked June profile, but was markedly increased in the post-stocking water column samples. The abundance of *tetA* and *int1* (ranging from 1.4 × 10^2^ to 3.7 × 10^5^ gene copy numbers) varied in the different sampling profiles, but the abundance of *tetA* was increased substantially in the December and February profiles. Levels of *dfrA1* and *bla*_TEM_ were much lower than the other ARGs (4.9 × 10^−1^ to 3 × 10^1^), and no clear temporal trend in the abundance of these genes was observed.

### 3.4. Correlation Analysis

Canonical Correspondence Analysis showed complex relationships between and among specific environmental parameters (temperature, pH, average precipitation), ARG abundance based on qPCR and bacterial community composition from amplicon sequencing analyses (Figure 7, Appendix A), of both FP-03 and Res water samples. A positive correlation was observed between targeted ARGs encoding resistance to antimicrobials used in FP-03 (*sul1*, *tetA* and *dfrA1*) and the class 1 integron integrase gene (*int1*; *p* < 0.01; *R* = 0.7282/*sul1*, 0.6322/*tetA*, 0.7945/*dfrA1*; *n* = 8 for each gene). Furthermore, positive correlation was observed between members of *Bacteroidetes* spp. in FP-03 water and ARGs *sul1*, *tetA* and *int1 (p* < 0.05; *R* = 0.7172/*sul1*, 0.869/*tetA*, 0.9279/*int1*; *n* = 8). Presence of *Flavobacterium* spp. was strongly correlated with *tetA* abundance in both water columns (*p* < 0.01; *R* = 0.9618/FP-3, 0.8819/Res; *n* = 8), and was positively correlated with *int1* in FP-03 water (*p* < 0.05; *R* = 0.755; *n* = 8). Similarly, *Fluviicola* spp. was positively correlated with *tetA* (*p* < 0.01; *R* = 0.8485; *n* = 8) and *int1* in FP-03 (*p* < 0.05; *R* = 0.8209; *n* = 8), whereas in Res water *Pseudomonas* spp were positively correlated with *tetA* (*p* < 0.01; *R* = 0.9171; *n* = 8). Cyanobacteria were positively correlated with temperature (*R* = 0.8011/FP-3, 0.8961/Res; *n* = 8), while relative abundance of Proteobacteria (*p* < 0.01; R = 0.836; *n* = 8) in FP-03 water was positively correlated with precipitation. At the genus level, *Microcystis* was positively correlated with temperature at both sites (*p* < 0.05; *R* = 0.8166/FP-3, 0.7163/Res; *n* = 8), while the opposite held for *Flavobacterium* (*p* < 0.05; *R* = -0.7584/FP-3, -0.8618/Res; *n* = 8).

## 4. Discussion

The objective of this study was to evaluate ARG and microbial community dynamics in a freshwater aquaculture pond for the duration of a full annual fish fattening cycle, following prophylactic administration of antimicrobials. Similar to previous studies that assessed ARG abundance in aquaculture systems [41,42], shotgun metagenomic analyses indicated that sulfonamide resistance genes were strongly correlated to antimicrobial use, as both *sul1* and *sul2* were relatively copious in FP-03 water columns, but almost completely absent in the non-stocked Res water columns. While *sul2* was abundant in all of the FP-03 profiles, the relative abundance *sul1* substantially increased subsequent to prophylactic application of trimethoprim/sulfomethoxazole at the beginning of the fish fattening cycle, suggesting that the relative abundance of bacteria harboring this gene increases upon exposure to sub-therapeutic sulphonamide concentrations. Correlations between prophylactic antimicrobial use in aquaculture and enhanced abundance of ARGs has been previously reported by Gao et al. [43] and Xiong et al. [1]. This phenomenon was supported by a previous study we conducted that linked prophylactic trimethoprim/sulfomethoxazole use in aquaculture ponds to sulphonamide resistance in *Aeromonas* spp. [12].

The ubiquitous presence of *sul*2 in all of the FP-03 profiles (including the pre-stocked Jul15 profile not exposed to antimicrobials) and its complete absence in the adjacent non-stocked Res, suggests that the FP-03 sediment may be a reservoir for sulphonamide-resistant bacteria. Previous studies have reported significant accumulation of ARGs in sediments impacted by aquaculture, which were subsequently characterized as potential ARG repositories [1]. Kobayashi et al. [44] and Suzuki et al. [45] reported similar augmented resistomes in Mekong river sediments impacted by aquaculture.

Contrary to sulphonamide resistance genes, we did not observe a clear correlation between prophylactic trimethoprim/sulfomethoxazole use and the propagation of trimethoprim resistance genes. Overall, there were significant discrepancies in the distribution and relative abundance of trimethoprim resistance genes in the water column samples depending on the type of analysis conducted. COSMOSID detected high abundances of *dfr*D in the July 13 Res water profile (but not elsewhere), whereas qPCR and CARD analyses reported *dfr*A and *dfr*E, respectively, in all profiles with no clear trend in relative abundance. These discrepancies can stem from a wide range of factors discussed below.

Class 1 integrons play an important role in carriage and dissemination of ARGs [46], and *sul*1 is often ubiquitous in these elements. The strong positive correlation between *sul1* and *dfrA1* and class-1 integron integrase genes (*int1*) observed in the qPCR analysis, suggests that these two ARGs are harbored on class-1 integrons. Nonetheless, more robust analyses specifically targeting integron gene cassettes [47] need to be conducted in order to confirm this assumption.

Results of both the qPCR and the COSMOSID analysis of metagenomic data showed increased abundance of tetracycline resistance genes in Dec15 and Feb16 FP-03 water samples (not reflected in the CARD metagenomics analysis). While oxytetracycline was applied to quarantined fish in some of the fishponds linked to FP-03, and has been used in quarantined fish in FP-03 in previous seasons, to the best of our knowledge, unlike trimethoprim and sulfomethoxazole, it was not used directly during this study. Therefore, it is probable that oxytetracycline concentrations in the water column are extremely low and most likely do not select for tetracycline resistance.

While the study provides strong evidence linking prophylactic use of trimethoprim/sulfomethoxazole to increased abundance of sulfonamide resistance genes, discrepancies between the relative abundances of *sul*1 and *sul*2, the various trimethoprim resistance genes and other ARGs between qPCR and the shotgun metagenomic analysis highlights the complexity of elucidating resistome dynamics in complex environments. Inconsistencies in relative abundance of ARGs detected using shotgun metagenomics rather than PCR have been reported previously [48]. Andersen et al. [49] suggested that they may stem from different detection sensitivity and potential inhibition associated with either of the methods, but concluded that shotgun metagenomics was more accurate in detecting spiked DNA within complex microbial communities. Collectively, we believe that the shotgun metagenomic analyses of the ARGs is more accurate than the qPCR, considering the potential biases associated with PCR amplification and the fact that primers can potentially miss a large fraction of the targeted ARGs.

Interestingly, we also found substantial inconsistencies between the CARD and COSMOSID outputs from the same metagenomic data. This is clearly associated with differences in both the ARG databases used for comparison and the stringencies of the algorithms used in the two annotation pipelines. The COSMOSID pipeline applies a highly curated database and positive hits are based on multiple targets, and therefore, it can be assumed that the results obtained using the COSMOSID pipeline are more reliable for identifying clinically relevant ARGs. Conversely, the CARD database is more exhaustive and includes a broader range of ARGs from environmental sources, and therefore, it may provide a more holistic overview of environmental resistomes.

When assessing ARG distribution and abundance in the analysed water columns, it is vital to consider the microbial community composition in addition to selective pressure. Certain phyla harbor specific ARGs and, therefore, fluctuations in ARG abundance can be dictated by shifts in bacterial community composition regardless of selective pressure. It is highly probable that the cyanobacterial blooms observed in the summer profiles influence microbial community composition and henceforth the scope and abundance of detected ARGs. Guo et al. [50] suggested that cyanobacterial blooms have an inhibitory effect on ARG abundance in freshwater microbiomes, but considering the above, it is not clear if this is a cause or effect.

Seasonal factors (i.e., temperature and radiation) were the primary drivers of the microbial communities in the FP-03 and Res water columns. However, significant differences in the microbiomes of the two water columns suggests other “fish-associated” factors such as oxygen saturation, organic load and residual concentrations of antimicrobials also play a role in defining the microbial community composition. Nakayama et al. [13] observed that the water column microbiota of different aquaculture systems are strongly dictated by aquaculture management practices. In one report, Fouladkhah et al. [51] stated that the rise in environmental temperatures facilitates an increase in infectious disease agents. The multifactorial complexity of aquaculture ponds makes it difficult to determine precisely which specific factors affect microbial community dynamics in the water column. Nonetheless, we were able to identify trends and compile hypotheses, based on associated metadata. For example, at the genus level, algal blooms in FP-03 predominantly comprised *Microcystis*, whereas Res water also contained high levels of *Synechococcus*. This finding is supported by previous reports indicating that *Microcystis* thrives in ecosystems rich in nutrients (such as intensive aquaculture ponds), whereas *Synechococcus* proliferates in more oligotrophic ecosystems [52,53].

*Sphingomonas* was positively correlated with *Microcystis* blooms, based on its higher relative abundance in August profiles from FP-03 as compared to the Res water column. This may be associated with the capacity of this genus to detoxify microcystins [54]. A previously study that investigated temperature effects on microbiomes of German lakes found that *Sphingomonadales* were an integral element of *Microcystis* sp. blooms, corroborating results from this study [55].

Specific screening of the shotgun metagenomes using the curated COSMOSID platform identified substantially higher loads of potential human and animal pathogens in the FP-03 water column relative to that of the Res. These included species belonging to *Mycobacterium*, *Aeromonas, Elizabethkingia, Klebsiella, Staphylococcus* and *Enterobacter*. Nonetheless, none of these strains were identified in more than one profile, and generally, they were detected at very low concentrations, suggesting that they may not have significant epidemiological relevance. This was true for *Aeromonas* and *salmonicida* (detected in the July profile), the causative agent of skin lesions in carp and koi [56], and a major pathogen in fishponds in the sampled region.

To the best of our knowledge, this is the first study to assess comprehensively the temporal microbiome and resistome dynamics of intensive aquaculture ponds that use antimicrobials. Although data are quite complex, we provide strong evidence that prophylactic use of trimethoprim/sulfomethoxazole in aquaculture systems facilitates increased relative abundance of sulfonamide resistance genes. Furthermore, we provide evidence of a highly dynamic microbial community in aquaculture ponds that are strongly influenced by both photosynthetic and heterotrophic bacterial communities in the water column. To fully understand the clinical risks associated with intensive aquaculture systems using antimicrobials, future studies should assess presence of other clinically-relevant ARGs in class 1 integrons and their association with mobile genetic elements.

## Figures and Tables

**Figure 1 microorganisms-08-01984-f001:**
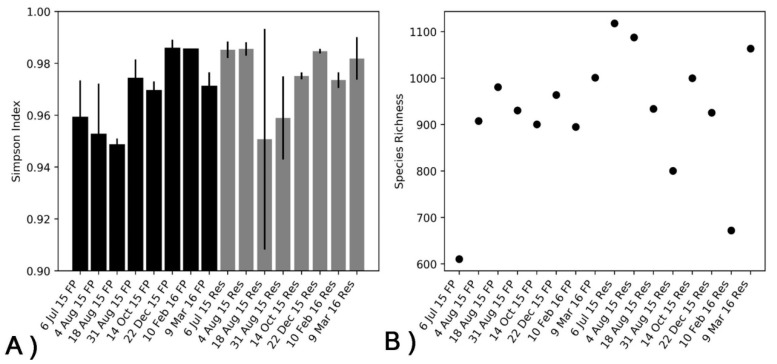
Temporal (alpha) diversity in the FP-03 (FP) and Res (Res) water columns at the Dor aquaculture research station based on analyses of 16S rRNA gene amplicon sequencing reads. (**A**) Simpson index and (**B**) Species Richness.

**Figure 2 microorganisms-08-01984-f002:**
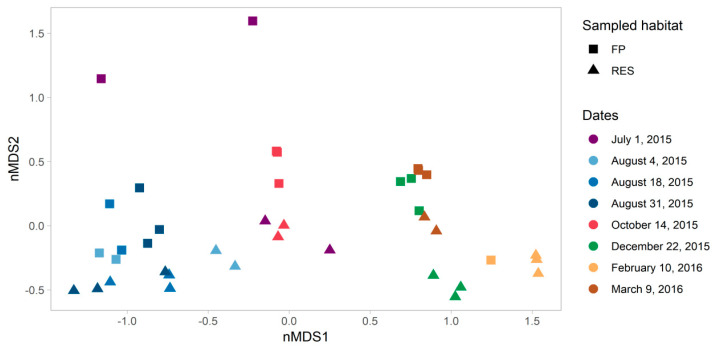
Non-metric multidimensional scaling (NMDS) biplot of a Bray–Curtis dissimilarity matrix of 16S rRNA gene amplicon sequencing data, showing the microbial community structure of 8 sampling points from FP-03 (FP- squares) and the Res (Res- triangles) at the Dor aquaculture research station. Stress = 0.06.

**Figure 3 microorganisms-08-01984-f003:**
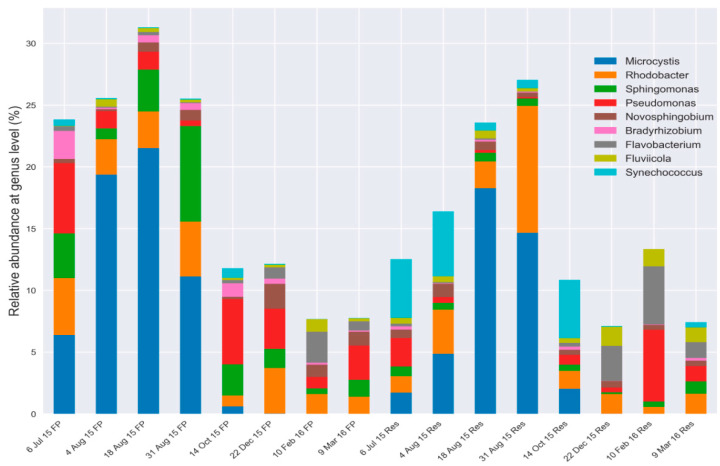
Relative abundance of predominant bacterial genera in FP-03 (FP) and the Res (Res) based on 16S rRNA gene amplicon sequencing analyses.

**Figure 4 microorganisms-08-01984-f004:**
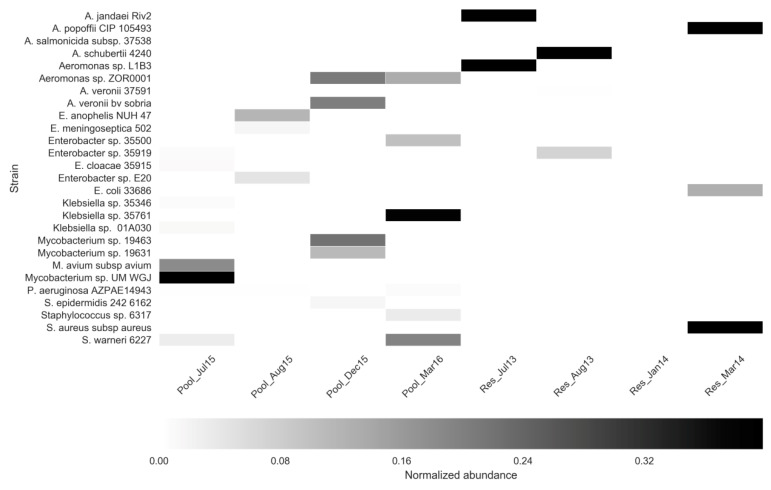
Heat map showing the relative abundance of predominant bacterial pathogens in FP-03 (Pool) and the Res (Res), based on shotgun metagenomic data analyzed using the highly curated COSMOSID pipeline.

**Figure 5 microorganisms-08-01984-f005:**
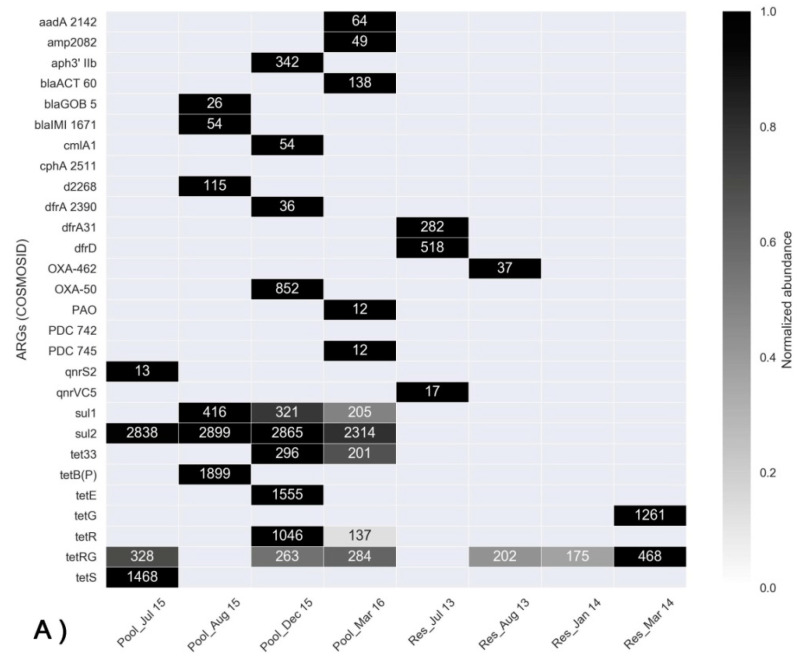
Heat maps showing the relative abundance of predominant ARGs in FP-03 (Pool) and the Res (Res) based on shotgun metagenomic data analyzed with the highly curated COSMOSID (**A**) and CARD (**B**) bioinformatic pipelines, respectively. Numbers indicate the absolute abundance of the targeted genes in the individual samples. Only ARGs potentially associated with mobile genetic elements were included.

**Figure 6 microorganisms-08-01984-f006:**
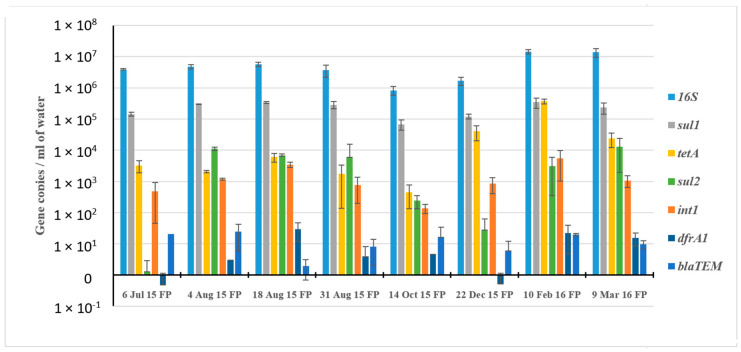
Abundance of selected qPCR-targeted ARGs in the FP-03 (FP) water column.

**Figure 7 microorganisms-08-01984-f007:**
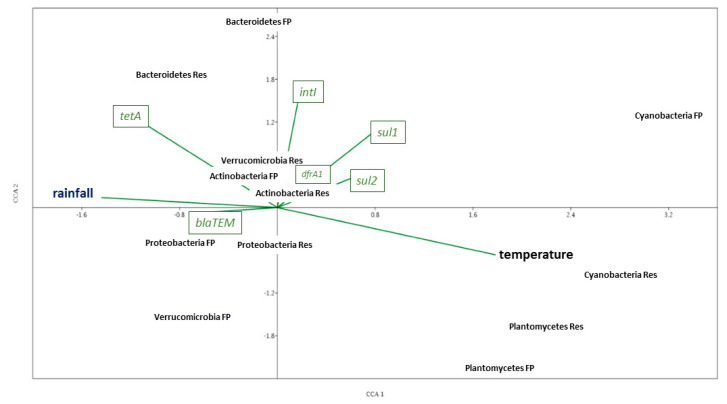
Canonical Correspondence Analysis (CCA) depicting the correlation between environmental factors (bold letters), ARGs (green text boxes) and prominent bacterial phyla.

**Table 1 microorganisms-08-01984-t001:** qPCR primers and conditions used in this study.

Gene Target	Primer Sequence	Amplicon (bp)	SYBR Green Master Mix Kit (Applied Biosystems)	Reference
*16S rRNA* (*CS1_341F* and *CS2_806R*)	5′-ACACTGACGACATGGTTCT ACANNNNCCTACGGGAGGCAGCAG-3′	195	FAST	[21]
	5′-TACGGTAGCAGAGACTTG GTCTGGACTACHVGGGTWTCTAAT-3′			
*sul1*	5′-CGCACCGGAAACATCGCT GCAC-3′	163	FAST	[34]
	5′-TGAAGTTCCGCCGCAAGG CTCG-3′			
*sul2*	5′-TCCGATGGAGGCCGGTATC TGG-3′	191	POWER	[34]
	5′-CGGGAATGCCATCTGCCTT GAG-3′			
*dfrA1*	5′-TTCAGGTGGTGGGGAGAT ATAC-3′	150	POWER	[35]
	5′-TTAGAGGCGAAGTCTTGG GTAA-3′			
*tetA*	5′-GCTACATCCTGCTTGCCT TC-3′	210	SELECT	[36]
	5′-CATAGATCGCCGTGAAGA GG-3′			
*bla_TEM_*	5′-TTCCTGTTTTTGCTCACCC AG-3′	113	SELECT	[37]
	5′-CTCAAGGATCTTACCGCTG TTG-3′			
*intI1*	5′-CCTCCCGCACGATGATC-3′	293	POWER	[38]
	5′-TCCACGCATCGTCAGGC-3′			

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
