# Peer review of "Temporal Resistome and Microbial Community Dynamics in an Intensive Aquaculture Facility with Prophylactic Antimicrobial Treatment"

_microorganisms, 2020, doi:10.3390/microorganisms8121984_

Round 1

Reviewer 1 Report

The revised submission is much improved, but an apparent typographical error has crept in: the resubmission is missing Table 1. This is because Table 1 in the original submission was renumbered to Table 2 and referred to as Table 2 in the text. This typo needs to be corrected.

Author Response

We apologize for leaving the table out. Table 1 has been included in the revised manuscript.

Reviewer 2 Report

I got the manuscript in the first round of the review process and I still feel that the manuscript is carefully conducted and that it is well written. The revisions requested by this reviewer are successively commented.

Author Response

There are no comments

Reviewer 3 Report

The authors considered most of suggestions indicated by a reviewer 1 and made some changes in whole manuscript. They filled gaps in methods and details about sampling. The manuscript was improved but still remains some text editing corrections.

In example: resistance genes to tetracyclines should be written as tet(A), tet(B)...

The authors claimed that they made changes in line 305 (former line 284). I can't see those corrections.

Author Response

General comments:

  1. In example: resistance genes to tetracyclines should be written as tet(A), tet(B)...

This study did not focus on specific tetracycline family resistance genes [such as tet(A) or tet(B)], but rather tetracycline resistance genes in general, and tet refers to the entire tetracycline resistance gene family as a whole. We believe this is legitimate.

  1. The authors claimed that they made changes in line 305 (former line 284). I can't see those corrections.

Line numbers might have been rearranged in the revised text. The change refers to the previous revision where “Typhimurium” ws changed to “typhimurium” (former line No.284 is no No. 299).